

# Sex-specific differences in symbiotic microorganisms associated with an invasive mealybug (*Phenacoccus solenopsis* Tinsley) based on 16S ribosomal DNA

Lu Wang[1], Xia Liu[1] and Yongming Ruan[1,2]

[1] Zhejiang Normal University, College of Life Sciences, Jinhua, Zhejiang, China
[2] Key Lab of Wildlife Biotechnology and Conservation and Utilization of Zhejiang Province, Jinhua, Zhejiang, China

## ABSTRACT

The ability of *Phenacoccus solenopsis* Tinsley (Hemiptera: Pseudococcidae) to utilize a wide range of host plants is closely related to the symbiotic bacteria within its body. This study investigated the diversity of symbiotic microorganisms associated with the sap-sucking hemipteran insect. Using deep sequencing of the 16S rDNA gene and subsequent analysis with the Qiime software package, we constructed a comprehensive library of bacterial operational taxonomic units (OTUs). We compared the microbial communities of female and male adult mealybugs. Our results showed significant differences in bacterial composition between the sexes, with Proteobacteria, Firmicutes, and Bacteroidetes being the dominant phyla in both female and male mealybugs. These results suggest that the diverse assemblage of symbiotic bacteria in *P. solenopsis* may be critical in enabling this insect to utilize a wide range of host plants by facilitating carbohydrate digestion and energy uptake.

## INTRODUCTION

Insects rely on bacterial symbionts for their nutritional ecology, as they help break down food or supply nutrients that are scarce or absent in the diet (*Feldhaar, 2011*; *Sharma, Malthankar & Mathur, 2020*). For example, some beetles with tyrosine-poor diets depend on endosymbionts to produce aromatic amino acids needed to synthesize their protective cuticle (*Dell'Aglio et al., 2023*). Many cases of insect symbiosis are obligate. Either the symbiont or the host insect cannot survive without the other. Some host insects have formed stable associations with pairs of bacterial symbionts that live in specialized cells and provide them with vital nutrients, such as aphids (*Munson & Baumann, 1993*) and termites (*Aanen et al., 2002*). A 2001 study reported that the Mealybug *Planococcus citri* (Hemiptera: Pseudococcidae) contains two bacterial symbionts in an unprecedented organization: an unnamed gamma proteobacterium, for which the name *Candidatus* Moranella endobia has been proposed (*McCutcheon & Von Dohlen, 2011*), lives within the beta proteobacterium *Ca.* Tremblaya princeps (*Von Dohlen et al., 2001*). However, later

Corresponding authors
Xia Liu, liux@zjnu.cn
Yongming Ruan, ruanym@zjnu.cn

research has shown that the secondary endosymbiont can infect the primary endosymbionts multiple times and even coevolve with their hosts (*Thao, Gullan & Baumann, 2002*). Previous research has reported an interdependent metabolic web in the nested symbiosis of mealybugs. It was put forward that the synthesis pathways of mealybugs for essential amino acids take place in Tremblaya and Moranella. Moreover, it was confirmed that Tremblaya and Moranella are the only bacteria present to any appreciable extent in the bacteriomes of mealybugs (*McCutcheon & Von Dohlen, 2011*; *Garber et al., 2021*).

The cotton mealybug, *Phenacoccus solenopsis* (Hemiptera: Pseudococcidae), was initially described in the USA in 1898 (*Tinsley, 1898*), suggesting that they are native to that country. The mealybugs feed on numerous crops, weeds, and ornamental plants. The adults and nymphs can severely damage leaves, fruits, main stems, and branches by feeding on phloem sap and excreting sugary honeydew (*Hodgson et al., 2008*; *Waqas et al., 2021*). Since it invaded India, Pakistan, and China, it has rapidly become a dominant pest of most of these plants. Research on the diversity of mealybugs is needed to understand how these pests survive and even infest the host plant since they cannot obtain sufficient amino acids but only carbohydrates from the plant sap. This involved symbionts that cohabited with the mealybug *P. solenopsis*. It was shown that the mealybug *P. citri* obtained amino acids from the beta-proteobacteria *T. princeps* and *M. endobia* (*McCutcheon & Von Dohlen, 2011*). However, whether other endosymbionts or bacteria in the mealybugs play a role in the other aspect is still unknown. This research explores the symbiont diversity in this pest. Studying symbiont diversity in mealybugs would better understand the potential relationship between symbionts and their host.

## MATERIALS AND METHODS

### Collection of samples

The mealybug, *P. solenopsis*, was initially collected from *Hibiscus syriacus* L. in Jinhua City (29°08′N, 119°37′E), Zhejiang Province in China in 2010, and was maintained on cotton (*Gosspium hirsutum* L.) in an insectary at 25–28 °C with a photoperiod of L12:D12 and 60–80% r. h. A single virgin female and thirty virgin male mealybugs were collected for DNA extraction. The virgin female was collected five days after eclosion, while virgin males were collected within two days after emergence (*Xiong et al., 2022*). Since the male mealybugs are too small to obtain accurate DNA with only one specimen, we collected thirty male mealybugs. These insects were then immersed in 99.9% alcohol and polished on parafilm.

### Construction of a bacterial 16S rDNA library

Insect DNA was isolated using the TaKaRa MiniBEST Universal Genomic DNA Extraction Kit (Takara, Shiga, Japan). PCR primers used a universal primer pair for the 16S rDNA V4 region (F: 5′-AYTGGGYDTAAAGNG-3′ and R: 5′-TACNVGGGTATCTAATCC-3′) and a fusion primer pair (F: 5′-index +AYTGGGYDTAAAGNG-3′ and R: 5′-TACNVGGGTATCTAATCC-3′). PCR products were purified using the TaKaRa DNA fragment purification kit. The interaction of 3′–5′

exonuclease and polymerase repairs the DNA fragment with the protruding ends. The products then add a base to the 3′ end of the DNA and assemble the junction. Enrich the DNA library fragment and test it using the method of Pico green and FLUORO to ensure quality. Libraries were then deep sequenced using Illumina Hiseq2000 according to the manufacturer's instructions.

## Analysis of the 16S rDNA library

Fragments of low quality and reads containing too many missing nucleotides were excluded based on the quality score. Each fragment must not contain six consecutive repetitive bases or fuzzy bases. After the basic dating process, using the tool Qiime (*Caporaso et al., 2010*) and depending on the sequence similarity, the OTU (operational taxonomic units) 0.97 was created. Depending on the method of tracing the most recent ancestor (*Morrow et al., 2008*), the information on the taxonomy of OTU was created. The Venn of the category of OTU 0.97 was drawn (*Knuth, 2006*). The rarefaction curves (*Gotelli & Colwell, 2001*; *Siegel, 2006*) were drawn to represent the sample depth visually. Then, the alpha diversity (*Whittaker, 1972*) based on the OTU 0.97 was calculated using the tool Mothur (*Schloss et al., 2009*) under the command 'Summary. Single' to display the diversity of symbionts intuitively. In addition, the rank abundance curve was constructed to show relative species abundance (*Solow & Polasky, 1994*; *Magurran, 2004*). Cluster the similar OTUs by phylum and genus, normalize each species to the same order of magnitude, and use log2 (M/F) (the abundance value 0 was replaced by 0.001) to count the differences of each taxon by phylum or genus. With the help of MEGAN4 software (*Huson et al., 2011*), a cladogram showing species abundance was constructed.

# RESULTS

## Length distribution of the sequenced 16S rDNA library

This study generated the 16S rDNA library separately from female and male mealybugs. The deep sequencing yielded 324,078 fragments (171,660 for the male insect and 152,418 for the female insect) and 322,561 high-quality fragments (170,810 for the male insect and 151,751 for the female insect). The distribution of sequence lengths showed a narrow length region, ranging from 223 to 228 bases (Fig. S1). These sequences must contain many repetitive elements of the 16S rDNA of the microorganisms. Furthermore, these repetitive elements could reflect the abundance of the insect's symbionts. Our result depended on the obtained 16S rDNA library to analyze the Mealybug's symbiont diversity. All original sequences were uploaded to NCBI's Sequence Read Archive (SRA) database. The SRA experiment accession is SRX365139, and the STUDY accession is SRP032536. The two original sequence runs have the accession SRR1013517 and SRR1013526.

## The creation of the OTU

From the Venn diagram of OTU 0.97 (Fig. 1), we can see that the number of species in male mealybugs is 851, the number of species in female mealybugs is 943, the number of species sharing male and female mealybugs is 373, the percentage of species sharing male and female mealybugs is 26.25%, and the total richness for all groups is 1,421.
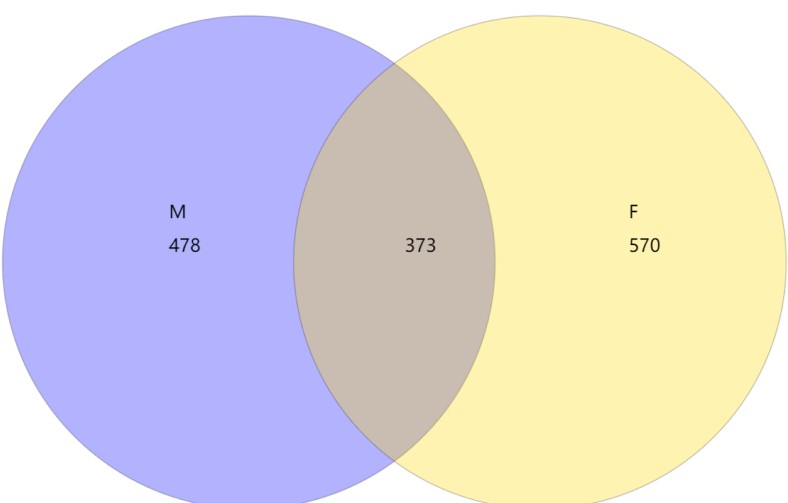

**Figure 1  Venn diagram of OTU of 0.97 of male (M) and female (F) mealybugs.**

## The analysis of species diversity and biodiversity indices based on OTU

Since the rarefaction curve (Fig. 2) flattens to the right, this implies that a reasonable number of individual samples were collected for OTU 0.97. More intensive sampling will likely yield fewer species. The rank abundance curve visually depicts relative species abundance (Fig. 3). For OTU 0.97, from which a reasonable number of individual symbiont samples were collected, the rank is up to 800. The relative abundance value approaches the X-axis unceasingly as the rank gradually increases to 800. This means that the OTU 0.97 is highly credible. The alpha diversity displays the diversity of symbionts of different sex of mealybugs in Table 1. Female mealybugs have a higher estimated richness (Chao and Ace) and a lower dominance (Simpson) than male mealybugs, but they also have a lower uncertainty (Shannon) and a lower evenness than male mealybugs. This suggests that female mealybugs have more OTUs but are more unevenly distributed than males. Both sexes' coverage values are very high, indicating that most OTUs have been observed in the samples.

## Analyze the differences in species abundance

The differences for the phyla and genera are presented in Table 2 and Table S1 (all counted values have three decimal places). The difference is significant if the log2 (M/F) value is greater than 1 or less than −1. Otherwise, the difference is not significant. The significant difference was marked in reseda (value less than −1) and pinky (value greater than 1) in Table 2, respectively. The difference between male and female mealybugs at the phylum level is insignificant in the richest Proteobacteria, and the difference in the second most abundant bacteria, Firmicutes, is significant.

Table S1 shows the analysis based on genera to understand better the differences between the microbial groups of female and male mealybugs. A total of 146 genera were classified by clustering the similar OTU on the genus. Most genera of microorganisms

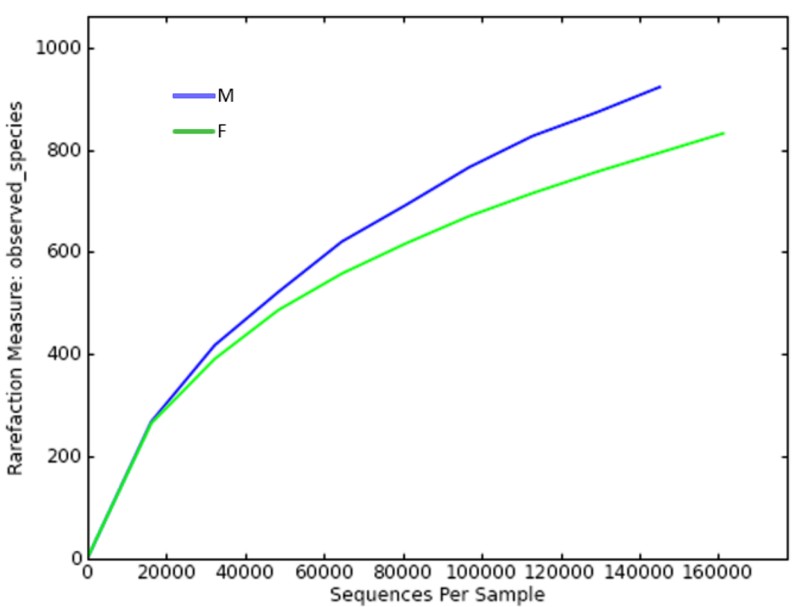

**Figure 2 Rarefaction curve of OTUs (similarity = 0.97) of male (M) and female (F) mealybugs.**

## Rank abundance curve

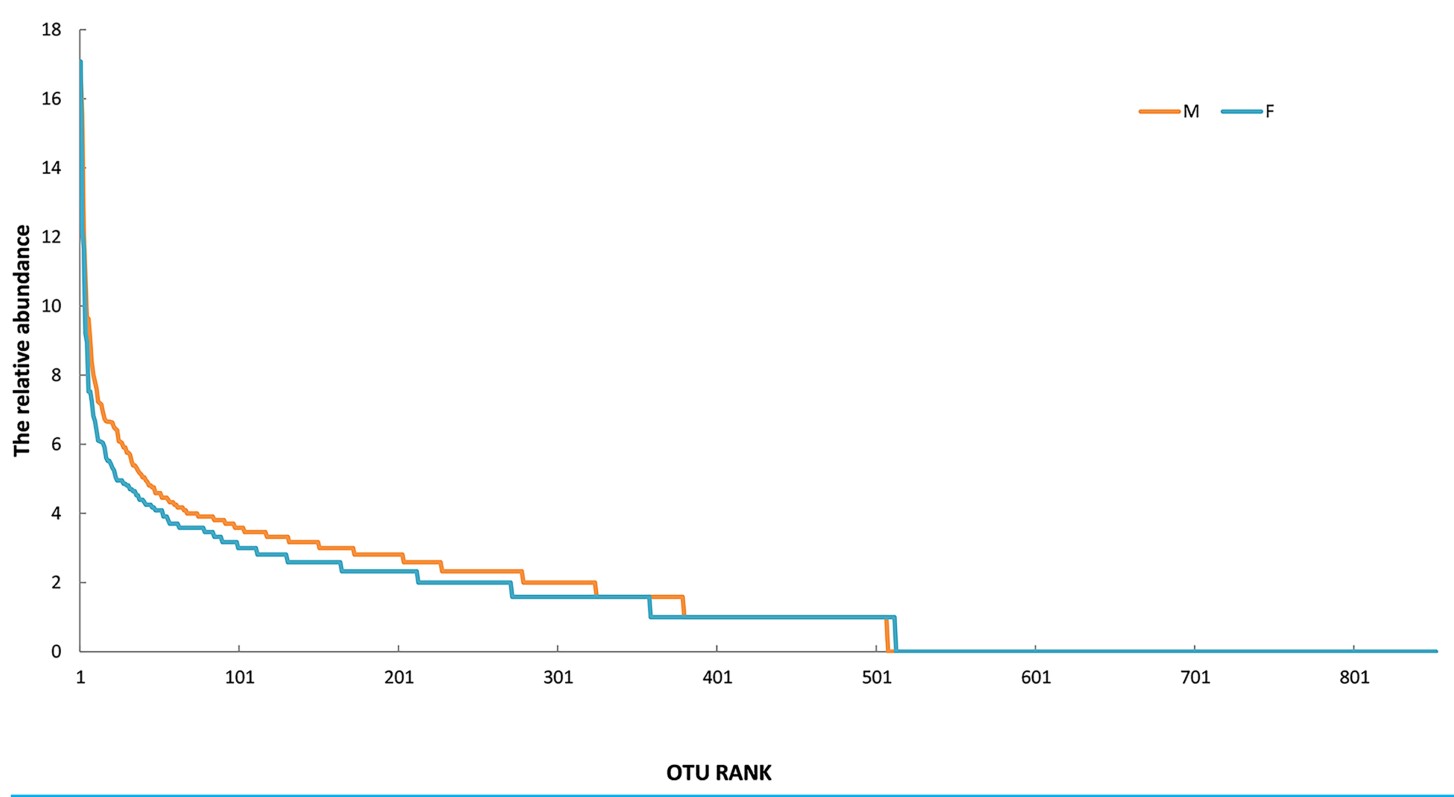

**Figure 3 The rank abundance curve of male (M) and female (F) mealybugs.**

**Table 1 The alpha diversity of OTU 0.97 of male and female mealybugs.**

| Sample | Chao | Ace | Simpson | Shannon | Coverage |
|---|---|---|---|---|---|
| Male | 1,308.33 | 1,505.15 | 0.4805 | 1.1887 | 0.9980 |
| Female | 1,540.84 | 1,497.30 | 0.8298 | 0.5958 | 0.9972 |

**Table 2 Phylum abundance of male and female mealybugs and difference time among them.**

| Phylum | Abundance | | Difference time log$_2$ (M/F) |
|---|---|---|---|
| | Male (M) | Female (F) | |
| Bacteria; Proteobacteria | 392,618.699 | 383,799.8 | 0.0327 |
| Bacteria; Firmicutes | 2,536.151 | 11,608.49 | −2.194 |
| Bacteria; Bacteroidetes | 1,166.209 | 1,294.225 | −0.150 |
| Bacteria; Actinobacteria | 967.156 | 253.046 | 1.934 |
| Bacteria; Cyanobacteria | 306.774 | 23.723 | 3.693 |
| Bacteria; Deinococcus-Thermus | 60.886 | 60.626 | 0.006 |
| Bacteria; TM7 | 23.418 | 0.001 | 14.5153 |
| Bacteria; Fusobacteria | 18.734 | 26.359 | −0.492 |
| Bacteria; Verrucomicrobia | 18.734 | 152.882 | −3.029 |
| Archaea; Euryarchaeota | 14.051 | 7.908 | 0.829 |
| Bacteria; Deferribacteres | 2.342 | 13.179 | −2.493 |
| Bacteria; Tenericutes | 2.342 | 5.272 | −1.171 |
| Bacteria; Acidobacteria | 0.001 | 2.636 | −11.364 |
| Bacteria; Lentisphaerae | 0.001 | 47.446 | −15.534 |

differed significantly between male and female mealybugs, and only 20 species did not differ significantly, including *Mycobacterium* (a genus of Actinobacteria that has its own family, *Mycobacteriaceae*), *Bacteroides* (a genus of Gram-negative bacilli), *Cloacibacterium* (established in sewage in Norman, Oklahoma, 2006). Future research should focus on the bacterium to confirm the host relationships of these tiny but significant bacteria in the Mealybug.

## Analysis of species abundance and primary bacterial genus

Species abundance was analyzed based on community structure. The pie charts of male and female mealybugs were based on the phylum (Fig. 4) and genus (Fig. 5), respectively. Regarding the phylum, the species diversity of the two sexes was identical. Proteobacteria (99% in male and 97% in female mealybugs) is the wealthiest phylum in female and male mealybugs, but there is no significant difference between female and male mealybugs (Table 2). The second most abundant phylum, Firmicutes, which is significantly different, was more abundant in female mealybugs than in male mealybugs. The content of other bacterial phyla is shown in Supporting Information (Table S2).

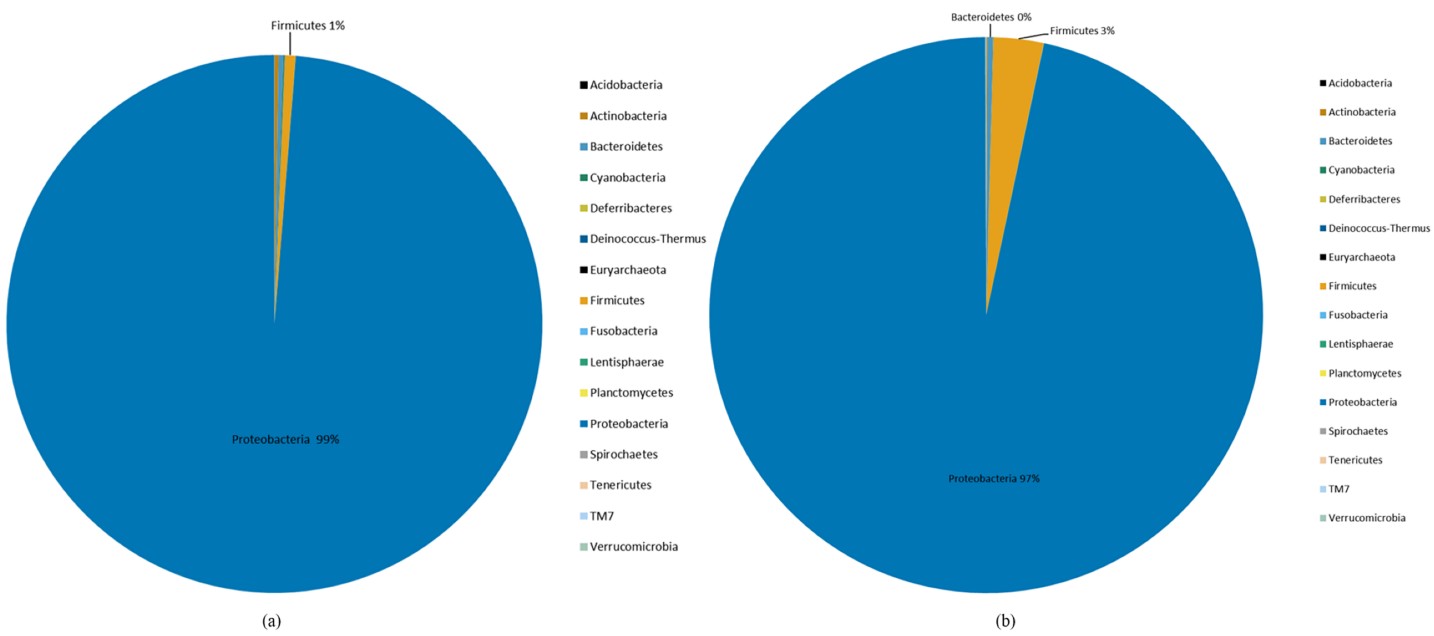

**Figure 4** The relative abundance of phylum distribution of male (A) and female (B) mealybugs.

The contents of the first 20 species of bacterial genera are listed in Table S3. It could be easily verified that the primary bacteria were the same in the two sexes. Regarding the genera, the wealthiest genus *Acinetobacter* significantly differs between male and female adult mealybugs. That is, the content in male mealybugs is higher than that in female mealybugs. Moreover, the second most abundant genus *Pseudomonas* is equally abundant in male and female mealybugs. However, the minute species are more numerous in female mealybugs than in males. These phenomena will be discussed later.

## System phylogeny analyses

Most of the sequenced 16S rDNA sequences aligned with the NCBI database were noted. The phylogeny based on these 16S rDNA sequences, including abundance distribution, was pieced to represent the content in male and female mealybugs concisely (Fig. 6). The richest proteobacteria include a variety of pathogens, such as *Escheriachia*, *Salmonella*, *Vibrio*, *Helicobacter*, and many other notable genera. In phylogeny, this phylum branches into six bacterial classes: Alphaproteobacteria, Betaproteobacteria, Gammaproteobacteria, Delta/Epsilon subdivisions, or Deltaproteobacteria, Mollicutes, and the genus *Treponema*. *Acinetobacter* is the most abundant bacterium in Gammaproteobacteria, Pseudomonadales, and Moraxellaceae. Most of the bacteria belonging to the phylum Proteobacteria are involved in the feeding of mealybugs. Some may even be involved in the reproduction of the insect (*Breeuwer & Werren, 1990*). The phylum Firmicutes includes four classes in this phylogeny: Bacilli, Clostridia, Erysipelotrichaceae, and Veillonellaceae. Bacteroidetes is the third most abundant phylum in both males and females and is divided into four classes: Bacteroidales, Cytophagaceae, Flavobacteriaceae, and Sphingobacteriales. These two bacterial strains are associated with obesity in humans (*Turnbaugh et al., 2006*).

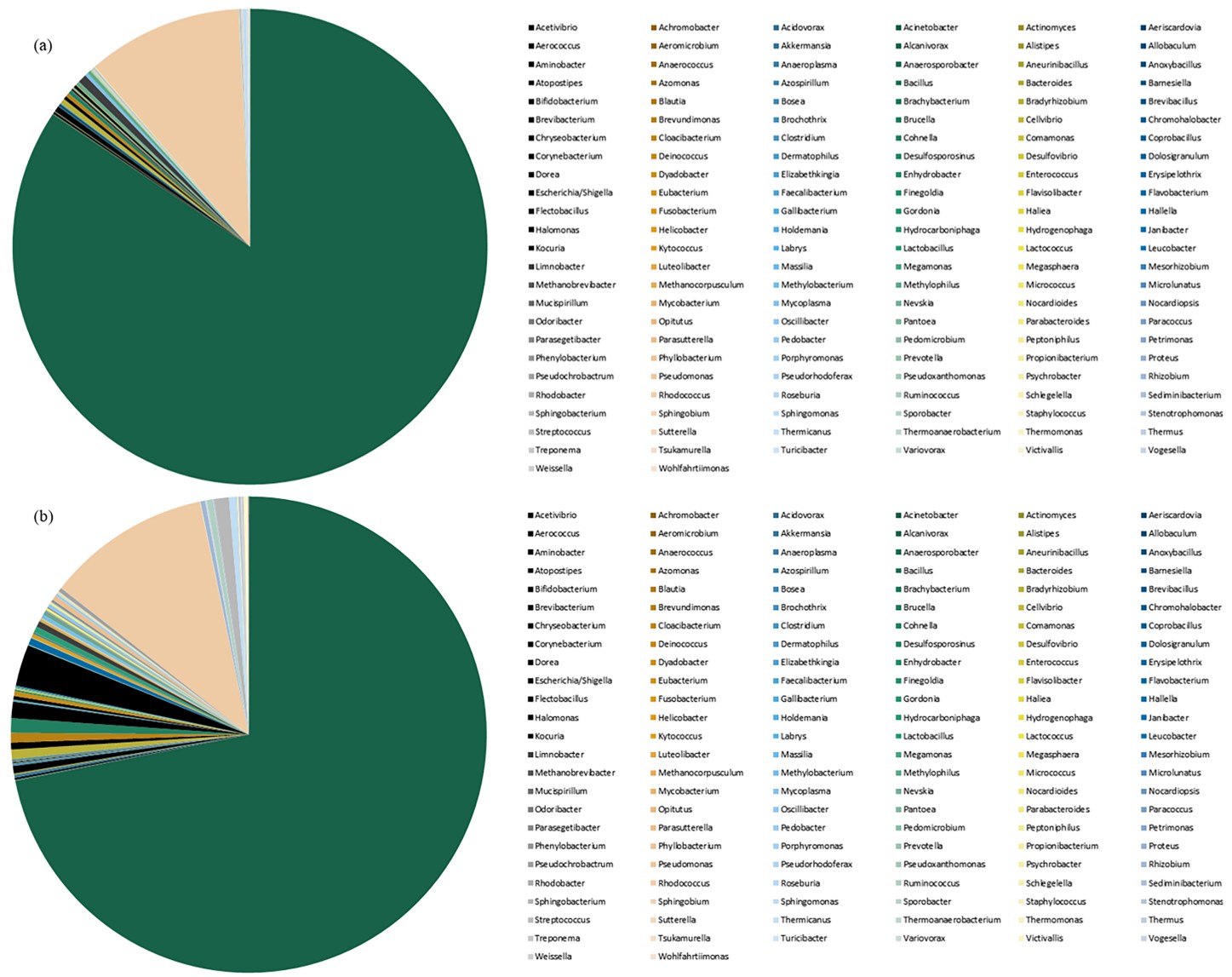

**Figure 5  The relative abundance of genus distribution of male (A) and female (B) mealybugs.**

## DISCUSSION

Mealybugs, like other phloem-feeding hemipterans, rely on bacterial endosymbionts to supplement their nutrient-poor diet (*Fan et al., 2022*; *Garber et al., 2021*; *Sharma, Malthankar & Mathur, 2020*; *Štarhová Serbina et al., 2022*). However, this inconspicuous pest, which feeds on only plant phloem sap, can rapidly spread and reach high population densities in an area. This ability must be strongly linked to the symbiont harbored within the mealybugs. The endosymbionts likely provide essential amino acids and other nutrients lacking in the mealybug's diet, allowing them to overcome nutritional deficiencies (*Garber et al., 2021*). The symbiont undoubtedly affects the habits or insect characters, just as it does in humans. In humans, depression, anxiety, and autism are
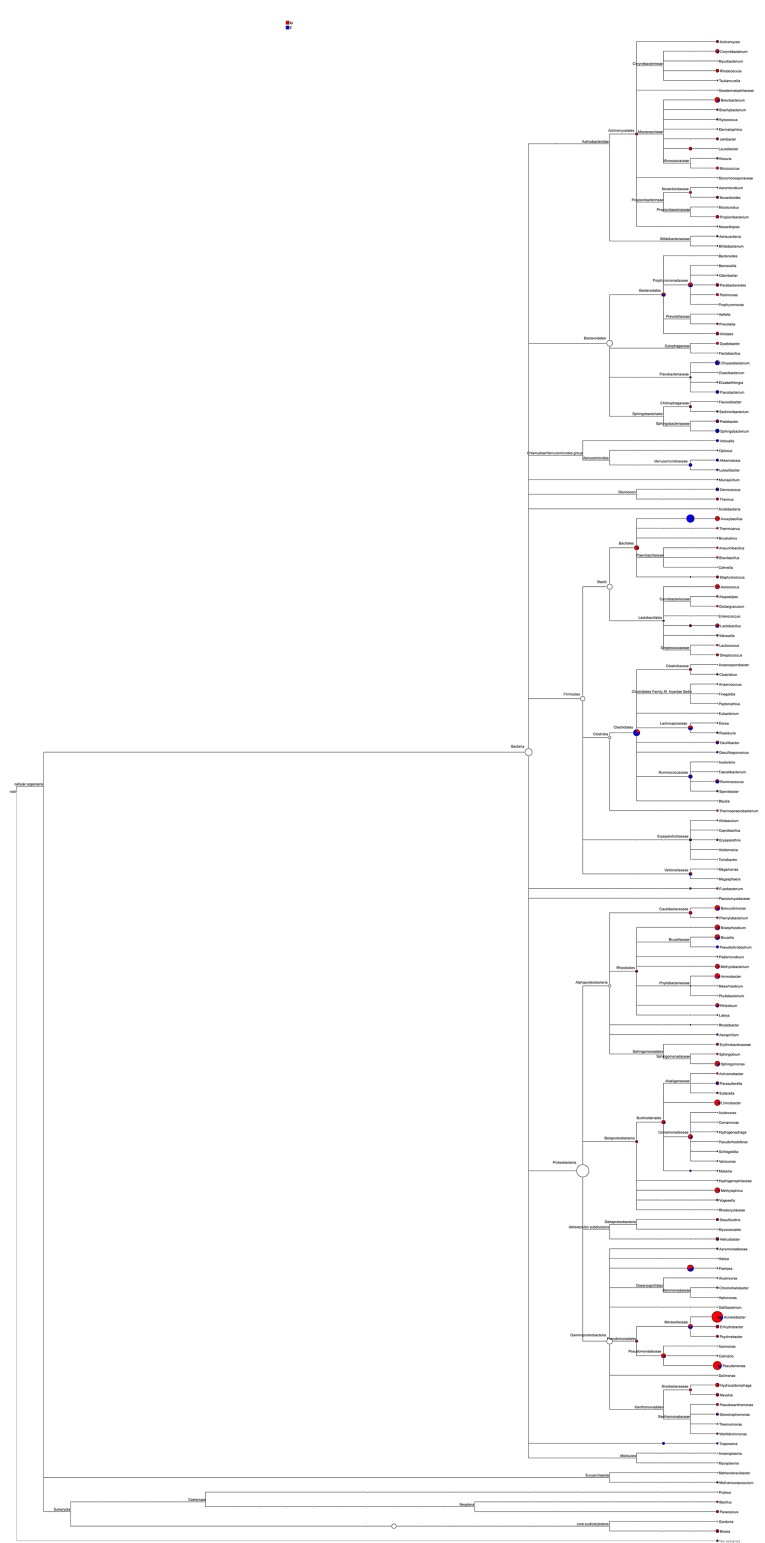

**Figure 6  Phylogeny and relative abundance of species detected in male and female mealybugs.**

associated with bacteria living in the body (*Fallon & Nields, 1994*; *Finegold et al., 2002*; *Mikocka-Walus et al., 2007*).

In addition to a nutritional benefit, endosymbionts may help mealybugs surmount plant defenses. For example, endosymbionts could detoxify plant allelochemicals and secondary metabolites that inhibit mealybug feeding or reduce fecundity. Plant-associated microbes enhance chemical and morphological defenses against herbivores, and endosymbionts may allow mealybugs to overcome such induced plant defenses (*Sharma, Malthankar & Mathur, 2020*). Infection with Rickettsia endosymbionts increased fertility, survival, and growth rate while decreasing development time in whiteflies (*Fan et al., 2022*). Likewise, mealybug endosymbionts could provide similar advantages in increasing population growth and adaptation. However, the complex relationships between endosymbionts, plant-associated microbes, and host plant defenses remain poorly understood and warrant further research. OTU analysis revealed that most bacteria in male and female adult mealybugs are significantly different. In the phylum, 75% of bacterial species are significant in different ways; this number is as high as 90% in the genus. The mealybugs possess several bacterial phyla, such as Proteobacteria, Firmicuter, Bacteroidetes, Actinobacteria, and Cyanobacteria. These five bacterial phyla are present in more than 99% of female and male mealybugs. A study reported that mealybugs partially harbour Rickettsia sp. (a type of Proteobacteria) (*Singh et al., 2013*), which may be essential for reproduction in female mealybugs (*Giorgini et al., 2010*). However, all reported symbionts were limited to the Proteobacteria phylum (most are α-Proteobacteria and β-Proteobacteria). Almost no research has reported other phyla of bacteria in mealybugs.

Many bacterial phyla have been revealed by using deep sequences to identify the diversity of symbionts in the Mealybug. For example, besides Proteobacteria, the phylum Firmicuter, Bacteroidetes, and several other phyla. In the past, Proteobacteria was associated with amino nutrition and the reproduction of mealybugs. In humans and other animals, Firmicuter and Bacteroidetes have been reported to be involved in obesity (*Turnbaugh et al., 2006*). These bacteria are also abundant in mealybugs, which may similarly aid host nutrition.

Moreover, obese people contain fewer Bacteroidetes than ordinary people. The Firmicutes distribute in the digestive tract of animals, and they can help the host to absorb excessive calories (*Singh et al., 2013*). This could be why Firmicutes are more abundant in female mealybugs than in males because females are more significant than males and need more calories to survive, especially when they need to spawn. Nevertheless, Bacteroidetes were less reported in the research. Its abundance was almost equal in female and male mealybugs. More research is needed to unravel the mystery, which must be done on the specific bacteroidetes genus.

Despite the limitation of a single female sample, this exploratory study revealed distinct trends in symbionts community differences between male and female adults of *P. solenopsis* that warrant further investigation. Follow-up studies with broader sampling across life stages, host individuals, and populations are needed to determine the generality of these patterns.

The interactions between mealybugs and endosymbionts are highly complex. Endosymbiont proliferation and persistence depend on host nutrient intake and genetic control (*Dell'Aglio et al., 2023*). However, endosymbionts provide nutrients and benefits to the host that are critical for growth, reproduction, and stress resistance. This interdependence has evolved relatively quickly, yet endosymbionts and hosts exhibit remarkable complementarity (*Garber et al., 2021*). Disentangling these multifaceted relationships will provide critical insights into how this ubiquitous agricultural pest has become so well adapted.

# CONCLUSIONS

This study explored the diversity of symbionts in the male and female adults of *P. solenopsis*. The results showed that most bacteria in male and female adult mealybugs are significantly different. Furthermore, the difference may be critical in enabling this insect to utilize a wide range of host plants by facilitating carbohydrate digestion and energy uptake.

## Funding

This work was supported by the Zhejiang Provincial Natural Science Foundation of China (No. LCN18C040001). The funders had no role in study design, data collection and analysis, decision to publish, or preparation of the manuscript.

## Grant Disclosures

The following grant information was disclosed by the authors:
Zhejiang Provincial Natural Science Foundation of China: LCN18C040001.

## Competing Interests

The authors declare that they have no competing interests.

## Author Contributions

- Lu Wang conceived and designed the experiments, performed the experiments, prepared figures and/or tables, and approved the final draft.
- Xia Liu conceived and designed the experiments, analyzed the data, prepared figures and/or tables, and approved the final draft.
- Yongming Ruan conceived and designed the experiments, analyzed the data, prepared figures and/or tables, authored or reviewed drafts of the article, and approved the final draft.

## DNA Deposition

The following information was supplied regarding the deposition of DNA sequences:
The sequences are available GenBank: SRX365139, SRP032536, SRR1013517, and SRR1013526.

## Data Availability

The raw measurements are available in the Supplemental Files.

## Supplemental Information

Supplemental information for this article can be found online at http://dx.doi.org/10.7717/peerj.15843#supplemental-information.

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
