# Peer review of "Sex-specific differences in symbiotic microorganisms associated with an invasive mealybug (Phenacoccus solenopsis Tinsley) based on 16S ribosomal DNA"

_PeerJ, doi:10.7717/peerj.15843_

## Round 0.1 · original submission · Major Revisions

Dear Drs. Wang and colleagues:

Thanks for submitting your manuscript to PeerJ. I have now received two independent reviews of your work, and as you will see, the reviewers raised some concerns about the research. Despite this, these reviewers are optimistic about your work and the potential impact it will have on research studying the biology of mealybug endosymbionts. Thus, I encourage you to revise your manuscript, accordingly, taking into account all of the concerns raised by both reviewers.

While the concerns of the reviewers are relatively minor, this is a major revision to ensure that the original reviewers have a chance to evaluate your responses to their concerns. There are many suggestions, which I am sure will greatly improve your manuscript once addressed.

Please include all of the relevant information in the Materials and Methods and figure legends to describe your analyses and interpretations (your work must be repeatable). There a few analyses that might improve your work, as suggested by the reviewers. Make sure the figures are clear (visulally).

Therefore, I am recommending that you revise your manuscript, accordingly, taking into account all of the issues raised by the reviewers. I do believe that your manuscript will be ready for publication once these issues are addressed.

Good luck with your revision,

-joe

Reviewer 1 ·

Basic reporting

This study investigated the diversity of symbionts in the male and female adults of P. solenopsis using deep sequencing of the 16S rRNA gene and subsequent analysis with the Qiime software package. Studying symbiont diversity in mealybugs would better understand the potential relationship between symbionts and their host. However, there are some issues in the manuscript that require further clarification.
Main points
The method description is unclear:
1. The studies of Xiong Zhenze (2022) and Bet Yuke (2021) et al, have shown that the endophytes of insects at different ages may vary. Please indicate whether the female and male insects were of the same age at the time of sample collection.
2. Line 90 “A single female and thirty male mealybugs were collected for DNA extraction.” This study did not specify the number of replicates and only used one female adult, which is not representative.
3. In the discussion section I suggest that some literature from the last two years could be added as support.
Other points:
1. Table 2 performs differential analysis at the "Phylum" level, are there any comparisons of differences at other taxonomic levels?
2. The title of the article is "16s rDNA", but the abstract, keywords and text are "16s rRNA", please revise to be consistent;
3. In Figure 5(b), the lines of the text annotation are crossed, please make it more beautiful for viewing.
4. "OTU" is misspelled as "OUT" in table 1 in the manuscript, please make the correction.
5. The font size of Figure 6 is too small and cannot be read clearly.

Experimental design

no comment

Validity of the findings

no comment

Additional comments

no comment

·

Basic reporting

The manuscript entiled "Sex-specific differences in symbiotic microorganisms associated with an invasive mealybug (Phenacoccus solenopsis Tinsley) based on 16S ribosomal DNA" reported theendosymbionst in male and female cotton mealy bugs. However the following concerns to be addressed.
Why 16S V3-V4 instead of whole 16S?
In result section I could observe more references, than discussion. In my opinion both result and discussion to be merged
How male and female malybugs were identified?
Why starvation was not given to insects?
Entire text 16s to be 16S
Phylogenetic figure is not visible
I suggest KEGG analysis of OTU
What are the bacterial phyla different from both male and female? Their exact role to be indicated

Experimental design

Please refer 1

Validity of the findings

Please refer 1

---

## Round 0.2 · accepted · Accept

Dear Drs. Wang and colleagues:

Thanks for revising your manuscript based on the concerns raised by the reviewers. I now believe that your manuscript is suitable for publication. Congratulations! I look forward to seeing this work in print, and I anticipate it being an important resource for groups studying mealybug biology and endosymbiont dynamics. Thanks again for choosing PeerJ to publish such important work.

Best,

-joe

·

Basic reporting

The manuscript has significantly improved in revised version. The authors answered almost all the queries. Fig 6 clarity has to be improved

Experimental design

-

Validity of the findings

It is scientifically sound

Additional comments

-